# Two-Pore Channels Regulate Expression of Various Receptors and Their Pathway-Related Proteins in Multiple Ways

**DOI:** 10.3390/cells10071807

**Published:** 2021-07-16

**Authors:** Sonja Grossmann, Robert Theodor Mallmann, Norbert Klugbauer

**Affiliations:** Institut für Experimentelle und Klinische Pharmakologie und Toxikologie, Fakultät für Medizin, Albert-Ludwigs-Universität Freiburg, Albertstr. 25, 79104 Freiburg, Germany; sonja.grossmann@pharmakol.uni-freiburg.de (S.G.); robert.mallmann@pharmakol.uni-freiburg.de (R.T.M.)

**Keywords:** two-pore channel, endolysosomal system, intracellular trafficking, receptor endocytosis, RNA sequence analysis

## Abstract

Two-pore channels (TPCs) constitute a small family of ion channels within membranes of intracellular acidic compartments, such as endosomes and lysosomes. They were shown to provide transient and locally restricted Ca^2+^-currents, likely responsible for fusion and/or fission events of endolysosomal membranes and thereby for intracellular vesicle trafficking. Genetic deletion of TPCs not only affects endocytosis, recycling, and degradation of various surface receptors but also uptake and impact of bacterial protein toxins and entry and intracellular processing of some types of viruses. This review points to important examples of these trafficking defects on one part but mainly focuses on the resulting impact of the TPC inactivation on receptor expression and receptor signaling. Thus, a detailed RNA sequencing analysis using TPC1-deficient fibroblasts uncovered a multitude of changes in the expression levels of surface receptors and their pathway-related signaling proteins. We refer to several classes of receptors such as EGF, TGF, and insulin as well as proteins involved in endocytosis.

## 1. Introduction

### 1.1. Genetic Diversity and Structure

Two-pore channels (TPCs) constitute a small family of cation channels within the membranes of endolysosomal organelles. In mice and humans, only two TPCs were identified, whereas several other species express three or even more functional tpc-genes. The diversity of tpc-genes, exon-intron structures, transcripts, and proteins and phylogenetic relationships was recently described [1,2,3]. The name “two-pore channel” suggests an ion channel structure with two separate pores, but, in fact, the channel protein consists of two subunits forming a homodimer. Each subunit comprises two homologous domains, and each domain consists of six transmembrane segments. This basic structure is well known from numerous classes of ion channels, and it is thought that TPCs form evolutionary intermediates between one-domain TRP and four-domain voltage gated Na^+^ or Ca^2+^ channels. The first 3D architecture was determined for a TPC from *Arabidopsis thaliana* using X-ray crystallography [4,5]. Later, this was followed by the 3D structures of mouse TPC1 and human TPC2 using electron cryo-microscopy [6,7]. Functional studies in combination with these structural data indicate that the first four transmembrane segments of each domain participate in voltage sensing, while the fifth and the sixth segments constitute the pore region [6,7].

### 1.2. Activation via NAADP and Putative NAADP-Binding Proteins

Initially, TPCs were identified as the long-sought class of ion channels opened by the second messenger nicotinic acid adenine dinucleotide phosphate (NAADP) [8]. NAADP is a very potent Ca^2+^ mobilizing messenger and was identified first in sea urchin eggs in 1995 [9]. Although TPCs participate in NAADP-mediated Ca^2+^ release from small acidic organelles, photoaffinity labeling studies failed to demonstrate a direct binding between NAADP and TPCs [10]. Instead, labeled NAADP bound to smaller proteins that may be part of a higher molecular complex that is associated with TPCs. This model would also explain earlier studies suggesting ryanodine receptors (RyRs) as putative NAADP targets by pointing to a common NAADP-binding protein that participates in different Ca^2+^ release mechanisms.

Recently, several groups presented their efforts in identifying NAADP receptors and introduced two promising candidates. Yan and Guan identified LSM12 as a putative NAADP interactor by showing that recombinant LSM12 bound to immobilized NAADP but not to NADP. These data were confirmed by using LSM12 knockout cells that demonstrated an abolished interaction between NAADP and TPCs and a reduced Ca^2+^ signal. Two other teams, Gunaratne with colleagues and Roggenkamp and coworkers, converged on JPT2 (Jupiter microtubule-associated homolog 2) as an NAADP binding protein [11,12]. The first group used a novel bifunctional “clickable” NAADP photoprobe to search for NAADP-binding proteins and identified the 23 kDa mammalian JPT2. JPT2 bound to NAADP with very high affinity, co-immunoprecipitated with TPC1, and was required for NAADP-induced Ca^2+^ release [11]. The second team isolated JPT2 by using purification studies with Jurkat cells. However, their coprecipitation and colocalization experiments with RyRs connects NAADP-signaling and JPT2 with this Ca^2+^ pathway and suggests an important role of the RyR/JPT2-complex for the first seconds of T cell activation [12].

### 1.3. NAADP and PI(3,5)P_2_ Affect Ion Permeability and Selectivity

Besides NAADP, the phospholipid PI(3,5)P_2_ was described as an endogenous activator of TPCs. A considerable number of studies characterized TPCs initially as non-selective Ca^2+^ -permeable channels activated by NAADP [1,8,13,14,15], whereas others described TPCs as highly-selective Na+ channels that are directly opened by PI(3,5)P_2_ [16,17,18,19]. For several years, these conflicting data remained unresolved, but recent work by Grimm and colleagues addressed this apparent contradiction [20,21]. They presented a model for TPC2 with two endogenous activators that result in distinct ion selectivity. A high throughput screen for membrane-permeable small molecule activators identified two distinct TPC2 agonists. One of them evoked robust Ca^2+^ signals and non-selective cation currents, whereas the other one induced only modest Ca^2+^ signals but a Na^+^ selective current. It turned out that the two agonists mimic the actions of NAADP and PI(3,5)P_2_, causing different ion permeability and selectivity of TPC2.

### 1.4. Function in the Endolysosomal System

Mammalian TPC1 and TPC2 are found in intracellular membranes of the endolysosomal system but not in the plasma membrane. Within these compartments, they show a smooth transition from TPC1 predominantly found in early and recycling endosomes to TPC2 mainly expressed in late endosomes and lysosomes [22,23,24,25]. TPCs contribute to the release of Ca^2+^ from these vesicles into the cytoplasm and thereby to a locally and temporally restricted increase of Ca^2+^, likely necessary for fusion and/or fission of endolysosomal membranes. This forms the basis for vesicle trafficking and sorting as a consequence for endocytic, recycling, and degradation processes [26,27]. Recently, the role of TPCs for phagocytosis was uncovered and compared with other Ca^2+^ channels [28]. In phagocytic cells, the NAADP/TPC pathway activates calcineurin, which in turn dephosphorylates and activates the GTPase dynamin-2.

TPCs also contribute to the uptake of Ca^2+^ in endolysosomal compartments and the filling of endoplasmic reticulum (ER) Ca^2+^ stores. Data from exocytosis experiments using mast cells suggested that TPC1 controls intracellular Ca^2+^ homeostasis and Ca^2+^ balance between the ER and the endolysosomes [29]. Thus, TPC1 deficiency disturbs Ca^2+^ homeostasis and granular histamine content and causes an altered filling of ER and organellar Ca^2+^ stores. In vivo, TPC1 knockout mice have fewer mast cells but are characterized by an enhanced exocytosis and mediator release due to an increased immunoglobulin-triggered Ca^2+^ signal [29].

In view of the importance of TPCs for intracellular trafficking, one might expect more severe phenotypes in existing knockout models. However, a closer inspection of these mouse models suggests that phenotypes come to light mostly under stimulated but not under basal conditions. This is evident in liver cells from TPC2 deficient mice; here, only mice fed with a high cholesterol diet demonstrated an accumulation of cholesterol in late endosomes, probably due to an impairment of endolysosomal fusion processes. Furthermore, TPC2 deficient mice were susceptible to cholesterol overload and liver damage only under this diet [27]. Another example is the IgE-triggered exocytosis observed in mast cells [29].

### 1.5. Function beyond Intracellular Trafficking

The question arises of the role of TPCs beyond involvement in endocytosis, recycling, and degradation. There is a huge number of cell surface receptors and transporters that are regulated via endo- or exocytosis or recycling and degradation. Furthermore, some receptors are still signaling even after endocytosis and as long as intracellular parts have access to the cytosol. Given the role of TPCs as general regulators of intracellular trafficking, one might expect effects of TPCs on surface expression and receptor signaling. These consequences should emerge in TPC-deletion studies, such as for TPC1, TPC2, and even more prominently in TPC1/2-double knockout cell lines. In this review, we first describe the role of TPCs for intracellular trafficking but mainly focus on the functions beyond those aspects. In particular, we used RNA sequence analysis from wildtype and TPC1-deficient fibroblasts to gain insight into the expression levels of surface receptors and their pathway-related signaling proteins.

## 2. Uptake and Trafficking of Bacterial Protein Toxins

Bacterial protein toxins were established as a tool to uncover the specific roles of different Ca^2+^ sources, i.e., classes of ion channels for determining individual endosomal trafficking routes [30]. These toxins elicit their effects after modification of intracellular target proteins in the host cells. They are taken up by receptor mediated endocytosis and hijack different endosomal routes to reach their final cytosolic destination. The toxins principally use two main routes to gain access to the cytosol: they can either be transferred from early or late endosomes (referred to as “short trip” toxins) or are transported all the way to the Golgi apparatus and the ER (“long trip” toxins). The translocation of the first group from early and late endosomes into the cytosol is driven by ongoing acidification. Diphtheria toxin (DT), the lethal factor of Anthrax toxin (LF/PA) and Pasteurella multocida toxin (PMT), are examples for this uptake route. The second group including Cholera toxin (CT) is retrogradely transported after endocytosis via the Golgi apparatus to the ER. To investigate the role of TPC1 for endosomal trafficking, we applied a set of “short trip” and “long trip” toxins as route-specific model substrates in wildtype and TPC1-deficient cells (MEF, HeLa, and J774 cells) [25].

The application of the “short trip” toxins to appropriate cell lines resulted in a delayed toxin uptake and reduction of target protein modification (PMT and DT). The cytotoxic effect of LF/PA in a TPC1-knockout macrophage cell line was also delayed. The result from the latter was confirmed by inhibition of TPC channels with tetrandrine, which decreased the effect of LF/PA in a dose-dependent manner. CT serving as example for a “long trip” toxin showed very little quantitative differences in cAMP accumulation in wildtype and TPC1-deficient cells within 3 h of toxin treatment [25]. However, another study that quantified fluorescently labeled CTxB in the Golgi apparatus found a slightly reduced accumulation of CTxB during a chase period of 2 h in TPC1-knockout fibroblasts [2]. The discrepancies in these observations may be caused by the variations in the observation periods and/or the different approaches. This study also highlighted the different roles of TPC1 and TPC2 for CT trafficking, which can be attributed to their distinct localization patterns in the endolysosomal system. In summary, the genetic inactivation of TPC1 reduced or attenuated the uptake and the toxic impact of all “short trip” toxins entering the host cells via early or recycling endosomes, whereas the effect of “long trip” toxins undergoing retrograde transport from late endosomes to the ER remained largely unchanged [25].

## 3. Uptake and Processing of Viruses

Another excellent model for investigating the roles of TPC1 and TPC2 for vesicle trafficking is based on endocytosis and endolysosomal processing of Ebola virus [31]. The virus causes a highly lethal and rapidly progressing hemorrhagic fever and is responsible for several severe outbreaks in Africa and remains a public health threat. Ebola viruses enter their host cells by macropinocytosis and are subsequently transported through the endolysosomal system, where viral glycoproteins are cleaved and bind to the Niemann-Pick C1 protein [32]. The final processing steps involve membrane fusion with lysosomes and release of the viral core into the cell cytoplasm, where replication starts. Sakurai and colleagues found that entry and trafficking of Ebola in HeLa and MEF cells require endosomal Ca^2+^ channels and identified TPCs as key players [31]. Thus, disrupting TPC function either by gene knockout, small interfering RNAs, or by pharmacological blockade prevented the escape of viral particles from the endosomal compartments into the cell cytoplasm. The alkaloid tetrandrine emerged as the most potent inhibitor in this study and was able to attenuate infection of macrophages, the primary target cells of Ebola, and demonstrated significant therapeutic benefit in a mouse model.

TPCs were also shown to be involved in controlling the life cycle of HIV [33]. A pharmacological inhibition or gene knockdown of TPCs attenuated the release of the HIV Tat protein from endolysosomes and transactivation of the long terminal repeat gene promoter. Obviously, this effect can be selectively attributed to TPCs since a knockdown of TRPML1 Ca^2+^ channels was without any impact [33].

More recently, there is accumulating evidence that the TPC complex is of critical importance for the infectivity of coronaviruses (MERS and SARS). For instance, an NAADP-mediated Ca^2+^ release promoted the activity of furin, a proprotein convertase necessary for internalization, trafficking, and release of MERS-CoV into the cytoplasm [34]. Later on, the same group demonstrated the importance of TPCs and their accessory protein JPT2 for the life cycle of SARS-CoV-2 [11]. Viral translocation trafficking was monitored in HEK293 cells using a whole set of inhibitors and siRNAs targeting TPCs and JPTs and resulted in a significant reduction of SARS-CoV-2 virus entry for each experimental condition that affected function of TPC1, TPC2, or JPT2 but not TRPML1 or JPT1. These data confirm not only contribution of TPCs for endolysosomal virus processing but also a role for JPT2 in a TPC complex.

In summary, above mentioned studies indicate that the development of pharmacological modulators of TPCs might be promising for the treatment of a range of viral infections.

## 4. Regulation of EGF-Receptor Expression and Pathway-Related Proteins

There is a huge number of transmembrane receptors whose signaling is regulated or terminated via receptor endocytosis, recycling, and/or lysosomal degradation. Among these, the epidermal growth factor receptor (EGFR) seems to be an ideal candidate to study the role of TPCs during these processes [35]. Although EGFR signaling is initiated after ligand binding at the cell surface, activated EGFRs are located within endolysosomal membranes after endocytosis, and EGFR signaling is ongoing [36,37,38]. Furthermore, activated receptors are internalized by clathrin dependent or independent endocytosis, and it was shown that ligand concentrations determine the preferential trafficking route of activated receptors [39]. At low EGF concentrations, uptake primarily occurs by clathrin mediated endocytosis, and EGFRs are transported via early and recycling endosomes back to the plasma membrane [40]. At higher EGF concentrations, increasing amounts of activated EGFRs are taken up by clathrin independent endocytosis and are routed via late endosomes to lysosomes for degradation [41]. In summary, these prerequisites form an ideal model to study the functions of TPCs for regulation of receptor endocytosis, recycling, and degradation and even for investigating downstream processes.

The deletion of TPCs in MEF cell lines resulted in an increased uptake of the EGF–EGFR complex and altered trafficking when cells were incubated with high concentrations of EGF. TPC1/2 double knockout cells accumulated even more EGF than single knockouts. Interestingly, the use of Rab5 as a marker of early endosomes uncovered that TPC2 and TPC1/2 double knockouts showed numerous examples of co-localization of Rab5- and EGF-positive vesicles, whereas wildtype and TPC1-deficient MEF cells exhibited only minor co-localization. These observations suggest that EGFR trafficking is more delayed in TPC2 knockout cells, which may be caused by a longer retention time of EGFR in late endolysosomal vesicles. Those data were collected using fluorescence microscopy and FACS-based approaches [35]. A similar effect was observed for LDL and its receptor since TPC2 deficient MEFs demonstrated a strong accumulation of LDL positive vesicles per cell following incubation with LDL-BodipyFL [27].

To further substantiate the role of TPCs for the EGFR network, the amount of total and surface accessible EGFR was quantified by Western blot analysis and EGF binding studies. For both approaches, the amount of EGFR was significantly increased in TPC single knockout cells but was highest in TPC1/2 double knockout cells. Additional experiments using cycloheximide as a protein synthesis inhibitor demonstrated that there were no significant differences in EGFR degradation kinetics between all genotypes and suggested that the higher expression levels of EGFR in TPC knockout cells cannot be attributed to an altered degradation process [35]. A similar approach also found a time and concentration dependent increase of EGF-positive vesicles in TPC2-deficient MEFs [27]. However, this study quantified EGFR levels in liver samples only by quantitative RT-PCR analysis but not at the protein level and did not find significantly increased EGFR transcripts in TPC2-deficient tissue.

Further studies were designed to discriminate between the two major regenerative mechanisms of EGFR availability, recycling, and de novo protein synthesis. As a result, from regenerative binding studies and quantitative RNA analysis, only EGFR de novo synthesis remains as the main mechanism responsible for the increased levels of surface accessible EGFR [35].

The results from the above mentioned experiments directly lead to a central question: what are the direct consequences of an increased EGFR expression for EGFR pathway related proteins? Initial evidence for quantitative changes in gene expression of EGFR pathway proteins following deletion of TPCs was obtained by an RNA sequencing analysis comparing wildtype and TPC1-deficient cells. The corresponding dataset indicated numerous quantitative changes in the gene expression levels of EGFR pathway proteins, whereby up- and downregulation were observed [35].

Based on these data, we propose a close link between function of TPCs and EGFR trafficking and signaling. EGFR dimerization and activation are initiated by EGF binding; subsequently, EGFR recruits downstream signaling complexes and triggers specific cellular signaling cascades. An important element is the ongoing EGFR signaling in intracellular organelles as long as the receptor-kinase domain is accessible from the cytosolic side [36,42]. During further maturation from early to late endosomes, EGFR is internalized in multivesicular bodies, and receptor signaling is stopped. Finally, receptor and ligand degradation occur in lysosomes. The deletion of TPCs causes a dysregulation of endolysosomal EGFR trafficking, a prolonged EGFR signaling, and an ongoing activation of downstream signaling pathways. Exactly this was found for phosphorylation of ERK1/2. In wildtype cells, pERK1/2 levels rapidly dropped down to initial levels, whereas, in TPC-deleted cells, they stayed at a high level for longer periods [35].

The question arises if further signaling pathways might be affected in the same way by inactivation of TPCs. Since *c-Jun* was shown to be a major factor for *Egfr* gene transcription, it is a promising candidate [43,44,45]. A quantification of phospho-c-Jun levels in wildtype and TPC-deficient MEF cells revealed elevated phospho-c-Jun levels in all cells lacking a functional TPC gene. Remarkably, this rise was already observable in the absence of EGF under unstimulated conditions. It is obvious that the increased *JNK* signaling is a major factor that contributes to the high EGFR expression found in TPC-deficient cells. We hypothesize that a prolonged EGFR signaling in endolysosomal signaling platforms caused by deletion of TPCs results in increased *JNK* signaling, which, in turn, leads to increased *Egfr* expression [35].

## 5. Regulation of Gene Expression via TPC1—RNA Sequencing Analysis

The massive effects of a TPC gene inactivation for the EGFR transportation network and for EGFR signaling lead directly to the investigation of EGFR-unrelated receptors and pathways. To achieve an unbiased approach, a detailed RNA sequencing analysis was performed as previously described [35]. Three independent untreated samples of wildtype and TPC1 knockout mouse embryonic fibroblast lines were included and compared. RNA sequencing pointed to substantially changed expression patterns in TPC1 knockout compared to wildtype cells. A total of 5255 genes were differentially expressed, with 2705 genes upregulated and 2550 genes downregulated. Remarkably, 1141 genes were upregulated and 647 were downregulated more than two-fold (Figure 1). The number of dysregulated genes due to loss of TPC1 gives a first hint at the relevance of efficient and well-regulated endolysosomal transport and trafficking for regulating gene expression.

Gene ontology (GO) enrichment analysis enables the search for over-represented GO terms in large biological data sets, for example, differential gene expression analysis results. Each gene product is annotated to one or more GO terms and can thereby be classified with respect to its molecular function (e.g., receptor binding or kinase activity), its cellular component (localization within the cell, for example, endosome or plasma membrane), and its biological processes (signal transduction or axon development, for instance) [47]. By screening for enriched terms, one can gain insight into alterations regarding all these aspects.

Gene ontology enrichment analysis using ClueGO (a Cytoscape software plugin, [48] revealed a very heterogeneous pattern of enriched terms regarding cellular components and molecular function. A pathway enrichment analysis using common pathway annotations (see legend to Figure 2; [47,49,50,51]) for the most up- and downregulated genes gave a similar picture. Interestingly, different pathways were not either simply up- or downregulated in total but seemed to be dysregulated with some highly upregulated members and others strongly downregulated (Figure 2).

We also considered if changes in the expression of genes involved in cell cycle regulation might indicate an inadequate synchronization of the cells in the biological replicates but could not confirm this eventuality. Furthermore, the small differences between replicates of one condition suggest that the changes in gene expression were not due to a lacking cell cycle synchronization or other experimental bias.

Some of the most significantly enriched pathways in our analysis were linked to downstream signaling of internalized receptors but also to processes involved in endocytosis. In particular, those were TGF-beta receptor signaling, MAPK signaling, regulation of IGF transport and uptake by IGFBPs and signal transduction (Figure 2). Interestingly, not only these classical pathways were affected but also cellular mechanisms involved in smooth muscle contraction or endochondral ossification.

Based on the data presented by Müller and colleagues, who described a close link between loss of TPC1 and gene expression changes in the EGFR pathway, we decided to have a closer look at pathways characterized by an ongoing signaling of internalized receptors, i.e., activated receptors present in “signaling endosomes”. Activated transforming growth factor beta (TGF-β) receptor was described for a continuous signaling in endosomal compartments (reviewed by [52,53,54]). Furthermore, the TGF-beta signaling pathway showed a highly significant enrichment in the GO analysis and takes center position in Figure 2.

## 6. Regulation of Genes Involved in TGF-β Receptor Signaling

The TGF-β receptor is a cell surface receptor kinase and is activated by a plethora of secreted polypeptides of the transforming growth factor beta superfamily. Members of the TGF-β superfamily are, among others, activins, bone morphogenetic proteins (BMPs), Nodal, and TGF-β1, -β2, and -β3. TGF-β signaling is involved in embryonic and adult control of proliferation, differentiation, and motility of cells or tissue. Genetic mutation, malfunction, or dysregulation of members of this pathway are associated with chronic inflammatory diseases, cancer, and fibrotic disorders [52,53].

Using RNA sequencing data [35], we studied transcriptional changes of TGF-ß receptor signaling pathway proteins. Expression data were transferred into a scheme highlighting the gene expression changes in TGF-β receptor signaling (Figure 3). Strikingly, the expression of TGF-β receptors was altered to a different extent. While expression of Tgfbr1 (TGF-β receptor 1) was almost halved (fold change 0.56) in TPC1-knockout MEF in comparison to wildtype cells, Tgfbr2 (TGF-β receptor 2) was more than quadrupled (fold change 4.31). This diverse picture was recognizable in downstream signaling proteins as well. A closer inspection indicated that expression of most of the downstream signaling proteins was reduced in TPC1-knockout cells. Furthermore, the reduction in expression was more pronounced than the increase of the few transcripts that were upregulated. Only six genes were upregulated more than two-fold, but 21 genes were downregulated more than two-fold (fold change 0.5).

## 7. Regulation of Genes Involved in Insulin Receptor Signaling

Thus far, we focused on receptors characterized by two features: (1) an altered expression in TPC1-deficient MEF cells and (2) an ongoing signaling in endosomal compartments following activation at the cell membrane. In contrast, the insulin receptor was not differentially expressed (0.95-fold in TPC1 knockout vs. wildtype cells, q > 0.05), and signaling in intracellular organelles is not described. Therefore, the insulin receptor is an ideal candidate to study the consequences of a TPC1 deletion for pathway-related proteins independent of the receptor expression level. In total, the expression of 55 transcripts related to insulin signaling was significantly changed in TPC1 knockout cells: 22 were up- and 33 were downregulated (Figure 4). Downstream proteins of insulin receptor were also affected by other receptors, particularly by receptor tyrosine kinases. This intertwining of pathways explains that even receptors that lack endosomal signaling are affected by deletion of TPCs and that their pathway related proteins can be differentially expressed as well.

## 8. Regulation of Genes Involved in Endocytosis

TPCs are important regulators of intracellular trafficking. Within the endolysosomal system, they show a differential distribution with TPC1 predominantly found in early and recycling endosomes and TPC2 mainly expressed in late endosomes and lysosomes [22,23,24,25]. Considering this distribution pattern, one might expect distinct effects of a TPC1 deletion on the transcript levels of proteins linked with endocytosis. In fact, the transcript levels of 91 proteins important for endocytosis were significantly changed in TPC1 knockout MEF cells (30 upregulated and 61 downregulated) (Figure 5). We categorized these proteins according to their function in the various parts of the endolysosomal system and according to their nature of endocytosis, i.e., whether they were clathrin dependent or independent.

Due to the dominant expression of TPC1 in early endosomes, the number of affected genes was highest in this compartment (18 affected early endosomal vs. eight late endosomal, eight recycling endosomal, and eight lysosomal proteins). In the context of clathrin-mediated endocytosis, five proteins were changed, and for clathrin-independent endocytosis, nine proteins were affected (Figure 5). The latter mechanism includes macropinocytosis, a form of endocytosis that is used by Ebola viruses. The substantial role of TPCs for uptake and processing of these viruses was extensively described [31].

## 9. Conclusions

TPCs are main regulators of intracellular trafficking. TPC1 is predominantly found in early and recycling endosomes, and TPC2 is mainly expressed in late endosomes and lysosomes. Therefore, genetic deletion of either TPC1 or TPC2 causes trafficking defects at various steps such as endocytosis, phagocytosis, recycling, or lysosomal degradation. These consequences emerge in all model systems investigated thus far: uptake of bacterial protein toxins, macropinocytosis and processing of some classes of virus, and endocytosis, recycling, and degradation of surface receptors in general. In this article, we present evidence that these defects are not restricted to trafficking only but go far beyond that. Of note, the use of cell lines in our studies restricts this general statement, but the concordant results provide an incentive to continue the work with appropriate in vivo models.

We focused on receptor signaling and on receptor-linked pathways by investigating the transcript levels of receptors and pathway related proteins. A detailed RNA sequencing of TPC1 deficient MEF cells uncovered massive changes in transcript levels when compared with wildtype cells. In particular, we observed transcriptional changes in pathway related proteins for receptors that are characterized by an ongoing endosomal signaling but also for receptors that are active exclusively at the plasma membrane. For the latter, we supposed that cross-linking of pathways explained the observed alterations. In the context of EGF receptor expression, *JNK* signaling was identified as a major factor that contributes to the high EGFR expression found in TPC-deficient cells. Significant changes were also found for clathrin dependent and independent endocytosis in TPC1 deficient cells, further corroborating an important role for TPC in different types of endocytosis.

## Figures and Tables

**Figure 1 cells-10-01807-f001:**
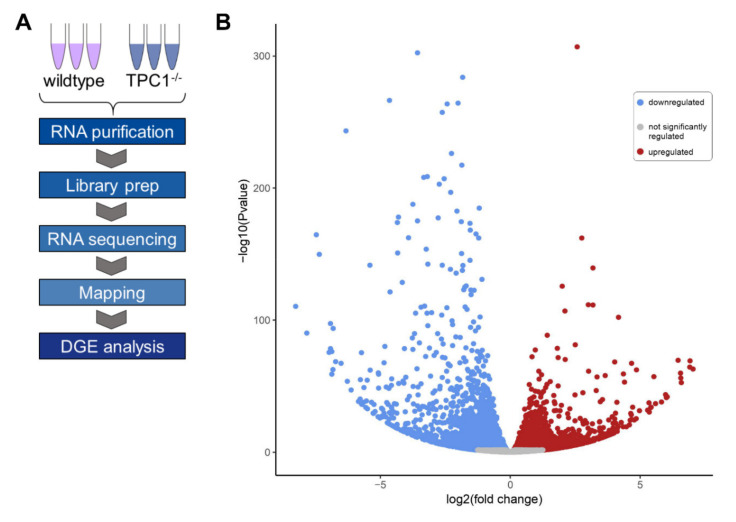
Differential gene expression (DGE) analysis. (**A**). Flow chart for RNA sequencing approach. (**B**). Volcano plot of RNA sequencing expression data of three independent samples (*n* = 3) of TPC1 knockout vs. wildtype mouse embryonic fibroblasts. Significantly up- and downregulated genes are indicated as blue and red dots, respectively. Grey area shows genes that were not differentially expressed. Only when p-value and false discovery rate were <0.05 gene expression change was considered statistically significant. RNA sequencing and differential gene expression analysis were performed as previously described [35]. Volcano plot was made using tools integrated in the Galaxy platform [46]. Original sequencing data were deposited in the Short Read Archive at the National Center for Biotechnology Information (NCBI) under the BioProject ID PRJNA694624.

**Figure 2 cells-10-01807-f002:**
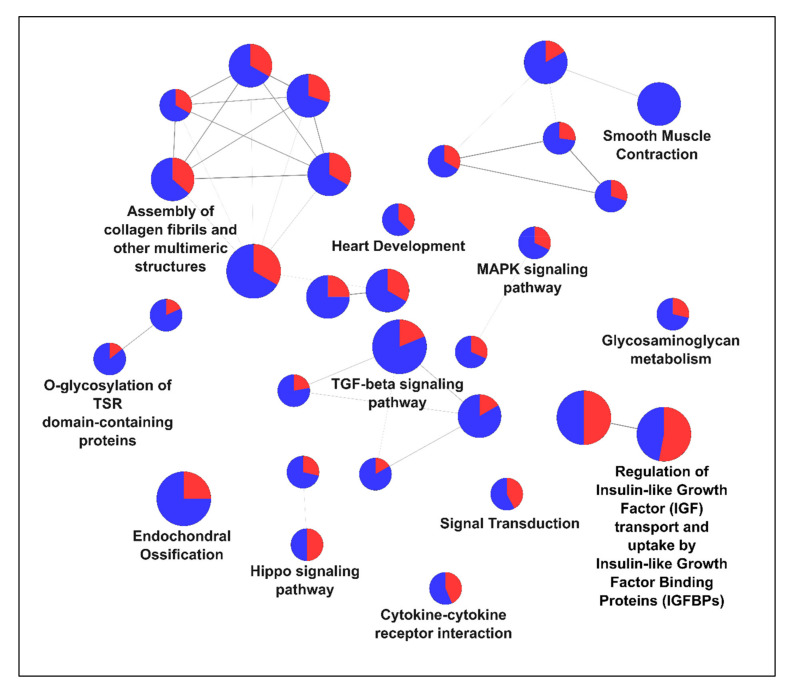
Pathway enrichment analysis of the 350 most up- and downregulated genes using ClueGO (Cytoscape software plugin, [48]. GO/Reactome/WikiPathways/KEGG pathway ([47,49,50,51]) functionally grouped networks with terms indicated as nodes (Benjamini–Hochberg *p* value < 0.05) linked by their kappa score level (≥0.4); only the label of the most significant term per group is shown. The size of the node correlates with the term significance. The node color shows the proportion of genes either upregulated (red) or downregulated (blue) associated with the term.

**Figure 3 cells-10-01807-f003:**
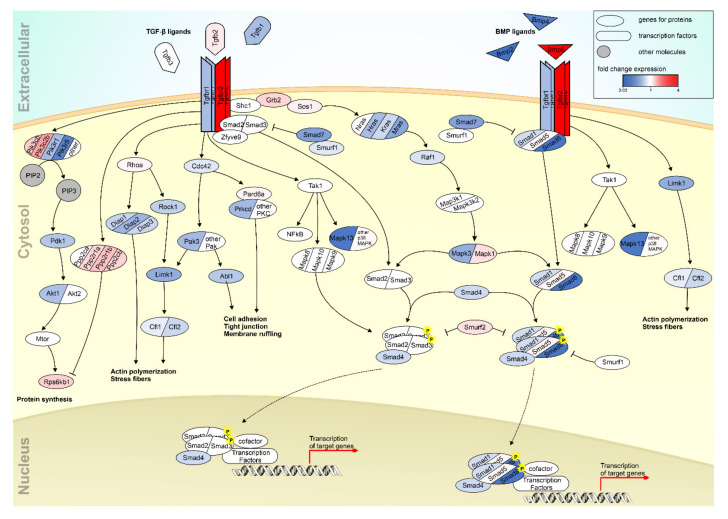
Differential gene expression of TGF-β receptor signaling pathway proteins in TPC1-deficient MEF cells compared with wildtype cells. Color code indicates changes in expression. Up- and downregulated genes are displayed in red and blue, respectively. Genes that show no significant differential expression (q > 0.05) in RNA-seq are shown in white (fold change = 1). Figure was adopted from an illustration and by courtesy of Cell Signaling Technology, Inc.

**Figure 4 cells-10-01807-f004:**
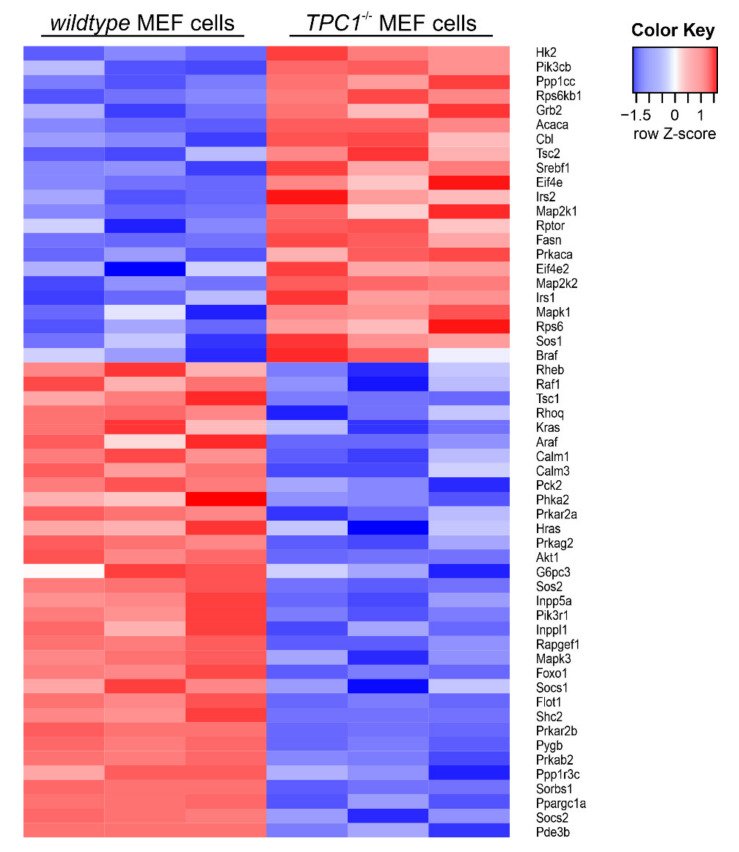
Heatmap of genes involved in insulin receptor signaling. Heatmap showing significantly differentially expressed genes (q < 0.05) in three independent samples (*n* = 3) of TPC1 deficient MEF vs. wildtype cells that are involved in insulin receptor signaling (KEGG pathway 04910; [49]. Each column stands for one independent sample of the depicted genotype. Expression values are shown as the Z-scores of the log2 transformed normalized counts for each gene. Red and blue are assigned to higher and lower expression, respectively, according to the color key in the upper right. Heatmap was created using tools integrated in the Galaxy platform [46]. Original sequencing data were deposited in the Short Read Archive at the National Center for Biotechnology Information (NCBI) under the BioProject ID PRJNA694624.

**Figure 5 cells-10-01807-f005:**
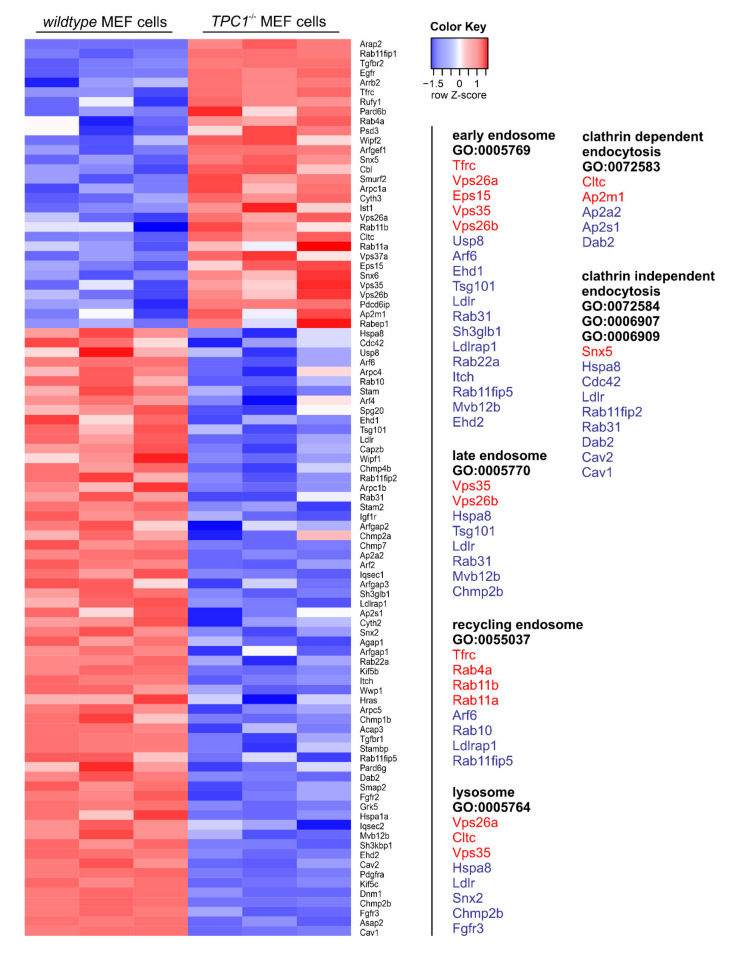
Heatmap of genes involved in endocytosis. Heatmap showing significantly differentially expressed genes (q < 0.05) in TPC1 deficient MEF vs. wildtype cells that are involved in endocytosis (KEGG pathway 04144; [49]. Expression values are shown as Z-scores of the log2 transformed normalized counts for each gene. Each column stands for one independent sample of the depicted genotype (*n* = 3). Red and blue are assigned to a higher and a lower expression, respectively, according to the color key in the upper right. Heatmap was created using tools integrated in the Galaxy platform [46]. All genes listed were compared to GO terms shown on the right. Only genes that were significantly changed were included and categorized according to the endolysosomal compartment or to the nature of endocytosis. Same color code was used as in heatmap. Original sequencing data were deposited in the Short Read Archive at the National Center for Biotechnology Information (NCBI) under the BioProject ID PRJNA694624.

## Data Availability

All sequencing data sets reported in this article are deposited in the Short Read Archive at the National Center for Biotechnology Information (NCBI) under the BioProject ID PRJNA694624.

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
