# Peer review of "Two-Pore Channels Regulate Expression of Various Receptors and Their Pathway-Related Proteins in Multiple Ways"

_cells, 2021, doi:10.3390/cells10071807_

Round 1

Reviewer 1 Report

General comments:
This review summarizes the effects of expression of two-pore channels (TPCs) on the expression of “various receptors”. Some sections of the review are written well, and some sections need revision as outlined in my comments. The source of the data used in heatmap figures is not clear. If the experiments are new, full details should be provided in the manuscript. Studies are mostly based on TPC1-deficient fibroblast cell lines. The text does not appear to relate to or provide evidence for the physiological and pathological significance of the findings. 

Specific comments:
    1. Abstract: All the sentences in future tense relating to the content of the review should be changed to past or present tense.
    2. The Introduction starts with the statements that “Two-pore channels (TPCs) constitute a small family of cation channels within the membranes of endolysosomal organelles. In mice and humans, only two TPCs have been identified, whereas several other species express three or even more functional tpc-genes”. Later in the Introduction and the rest of the manuscript (MS) the authors keep mentioning TPC1 and TPC2 without noting the relationships, homologies between these proteins.
Since this is a review manuscript, the authors should include a Table(s) summarizing the basic properties of the genes and the encoded proteins. The table(s) should include the following headings: Gene symbol, chromosome, Uniprot and/or Ensembl IDs, length of the genes (can include number of exons and introns), protein length in aa's and Mass (Da). These data should be provided for at least Homo sapiens and Mus musculus. If the gene(s) encodes biologically significant isoforms encoded by alternative transcripts, the differences in the properties and expression patterns could be summarized in an additional Table. The gene and protein names and abbreviations used should follow gene nomenclature recommendations.
Tissue-specific expression of the genes may be summarized in an additional Table.
    3. Grammar problems: Continuing my comment about the Abstract, I see similar problems about the tense of the sentence in the Introduction. For example, in line 39 the authors write that “Initially, TPCs have been identified as the long-sought class of ion channels”. This sentence should be written in past tense and not in the present perfect tense. There are more errors of this type. The whole MS should be re-read and revised.
    4. The Introduction mentions a potpourri of subjects including second messengers of TCPs, receptors of NAADP, activators of TCP, intracellular location of TCPs, ion selectivity of TCPs, and the role of TPCs in endocytosis, recycling and degradation. This presentation is unfocused and confusing. I suggest selecting a few subjects and presenting these in detail under specific subtitles.
    5. Line 79: Here the authors note that “Mammalian TPC1 and TPC2 are found in intracellular membranes of the endolysosomal system, but not in the plasma membrane.” Could you provide confocal microscopic images of the localizations mentioned here?
    6. Line 182: Change “Among them” to “Among these”.
    7. Line 184: Delete “all”.
    8. It appears that most if not all the studies reviewed were based on cultured cell lines such as MEF cells. In some sections, such as Section 3, results of experiments are mentioned but the cells in which these studies were carried out were not mentioned. For all studies, the cell system in which the experiments were carried out should be mentioned. The conclusions should also include a sentence about the cell systems in which the studies were carried out. If there are no studies in normal in situ, in vivo systems then a reservation should be added that the conclusions are based on artificial cell systems.
    9. Line 261: For RNA sequencing analysis, the authors note here that “Three independent untreated samples of wildtype and TPC1 knockout mouse embryonic fibroblast lines were included…” This statement explains the three sample columns in Heatmap figures. Therefore, this should be mentioned also in the legend of heatmap figures.
    10. Line 327: What is the source of “RNA sequencing data”? If the source is a previous study, it should be cited. If not, the full experimental details should be provided in this manuscript.
    11. Line 329: Here and throughout the MS, all instances of lowercase “(figure. ..) should be capitalized.
    12. Line 376: Change “Due to its dominant expression of TPC1” TO “Due to the dominant expression of TPC1”.
    13. Lines 377-379: Numbers less than ten should be written as text rather than as numbers. Ex.: Write five instead of 5.

Author Response

General comments:
This review summarizes the effects of expression of two-pore channels (TPCs) on the expression of “various receptors”. Some sections of the review are written well, and some sections need revision as outlined in my comments. The source of the data used in heatmap figures is not clear. If the experiments are new, full details should be provided in the manuscript. Studies are mostly based on TPC1-deficient fibroblast cell lines. The text does not appear to relate to or provide evidence for the physiological and pathological significance of the findings. 

We have added information on the source of the data in the legends to figures and also at the end in “data availability”.

In “Conclusions” we added a remark that points to the fact that the RNA sequencing data and effects on receptor signalling are based on MEF cells and not on in vivo models.

Specific comments:
    1. Abstract: All the sentences in future tense relating to the content of the review should be changed to past or present tense.

We apologize for the use of different tenses and corrected these sentences.

  1. The Introduction starts with the statements that “Two-pore channels (TPCs) constitute a small family of cation channels within the membranes of endolysosomal organelles. In mice and humans, only two TPCs have been identified, whereas several other species express three or even more functional tpc-genes”. Later in the Introduction and the rest of the manuscript (MS) the authors keep mentioning TPC1 and TPC2 without noting the relationships, homologies between these proteins.
    Since this is a review manuscript, the authors should include a Table(s) summarizing the basic properties of the genes and the encoded proteins. The table(s) should include the following headings: Gene symbol, chromosome, Uniprot and/or Ensembl IDs, length of the genes (can include number of exons and introns), protein length in aa's and Mass (Da). These data should be provided for at least Homo sapiens and Mus musculus. If the gene(s) encodes biologically significant isoforms encoded by alternative transcripts, the differences in the properties and expression patterns could be summarized in an additional Table. The gene and protein names and abbreviations used should follow gene nomenclature recommendations.
    Tissue-specific expression of the genes may be summarized in an additional Table.

We intensively discussed this point in our group and also with the Guest Editor of the special issue. We think that the insertion of such a detailed information on the genes, transcripts and proteins etc. of TPCs in tabular form does not fit into the scope of our manuscript. Instead, we have added several references of basic publications that summarize this data.

  1. Grammar problems: Continuing my comment about the Abstract, I see similar problems about the tense of the sentence in the Introduction. For example, in line 39 the authors write that “Initially, TPCs have been identified as the long-sought class of ion channels”. This sentence should be written in past tense and not in the present perfect tense. There are more errors of this type. The whole MS should be re-read and revised.

We apologize for the use of different tenses and revised the whole MS accordingly.

  1.   The Introduction mentions a potpourri of subjects including second messengers of TCPs, receptors of NAADP, activators of TCP, intracellular location of TCPs, ion selectivity of TCPs, and the role of TPCs in endocytosis, recycling and degradation. This presentation is unfocused and confusing. I suggest selecting a few subjects and presenting these in detail under specific subtitles.

The aim of our review is a presentation of the most recent literature in this field. Therefore, at first glance the choice of topics in the introduction seems to be arbitrary, but a closer look to the selected topics indicates that we follow a clear logic. After reviewing basic points such as genes and structure, we continue with the activation mechanisms (NAADP and PI(3,5)P2) and present the most recent findings in this area (identification of putative binding proteins, role of activators for ion selectivity). Finally, we discuss the function in the endolysosomal system and lead to the main part of our MS. To avoid the impression of a random selection of topics, we added subtitles in the introduction.

  1.   Line 79: Here the authors note that “Mammalian TPC1 and TPC2 are found in intracellular membranes of the endolysosomal system, but not in the plasma membrane.” Could you provide confocal microscopic images of the localizations mentioned here?

In the MS we refer to four papers in this context. In our lab we focussed on the expression of TPC1 and present its localization within the endolysosomal system in Castonguay et al., 2017. To our knowledge, this is the most detailed presentation by using a set of eight intracellular markers. We do not have additional images that would allow a deeper insight.  

  1. Line 182: Change “Among them” to “Among these”.

Thank you for drawing our attention to this point, it has been corrected.

  1. Line 184: Delete “all”.

Thank you for drawing our attention to this point, it has been corrected.

  1. It appears that most if not all the studies reviewed were based on cultured cell lines such as MEF cells. In some sections, such as Section 3, results of experiments are mentioned but the cells in which these studies were carried out were not mentioned. For all studies, the cell system in which the experiments were carried out should be mentioned. The conclusions should also include a sentence about the cell systems in which the studies were carried out. If there are no studies in normal in situ, in vivo systems then a reservation should be added that the conclusions are based on artificial cell systems.

We re-read our manuscript accordingly and added the information for the cell lines in the text. Nevertheless, the information for the cells was already indicated in the legend to figures. We extended the “Conclusions” and added the reservation that our studies rely on MEF cells, but not an in vivo system.

  1. Line 261: For RNA sequencing analysis, the authors note here that “Three independent untreated samples of wildtype and TPC1 knockout mouse embryonic fibroblast lines were included…” This statement explains the three sample columns in Heatmap figures. Therefore, this should be mentioned also in the legend of heatmap figures.

Thank you for drawing our attention to this point, it has been corrected.

  1. Line 327: What is the source of “RNA sequencing data”? If the source is a previous study, it should be cited. If not, the full experimental details should be provided in this manuscript.

We added the reference of our previous study (Muller et al 2021) and also included the source of the data in the legends to figures and at the end in “data availability”.

  1. Line 329: Here and throughout the MS, all instances of lowercase “(figure. ..) should be capitalized.

Thank you for drawing our attention to this point, it has been corrected.

  1.   Line 376: Change “Due to its dominant expression of TPC1” TO “Due to the dominant expression of TPC1”.

Thank you for drawing our attention to this point, it has been corrected.

  1.   Lines 377-379: Numbers less than ten should be written as text rather than as numbers. Ex.: Write five instead of 5.

Thank you for drawing our attention to this point, it has been corrected.

Reviewer 2 Report

Line 4: authors have no associated affiliations

Line 9: “They have been shown to  provide transient and locally restricted Ca2+-currents, likely  responsible for fusion and/or fission events of endolysosomal membranes and thereby for intracellular vesicle trafficking.” The role as Na+ channels is not considered in an adequate manner.

Line 38: A reference is needed.

Line 70: The reference Lagostena et al (doi: 10.1038/srep43900) should be included in this list.

General remark: It should be discussed how one can reconcile the fact that TPC1 and TPC2 KO mice have such a mild phenotype if TPC channels are so important for trafficking.

Author Response

Comments and Suggestions for Authors

Line 4: authors have no associated affiliations

We added the superscript.

Line 9: “They have been shown to provide transient and locally restricted Ca2+-currents, likely responsible for fusion and/or fission events of endolysosomal membranes and thereby for intracellular vesicle trafficking.” The role as Na+ channels is not considered in an adequate manner.

We agree with the reviewer that permeability for Na+ is a very important issue. We addressed this topic in the introduction and refer to a multitude of literature describing the conflicting data concerning ion permeation and selectivity. We also refer to the work of Grimm and colleagues, who identified activators that mimic the action of either NAADP or PI(3,5)P2 and who present a model explaining the different ion permeability and selectivity. The complexity of the entire issue is considerably high and is better addressed in the “Introduction” instead of the “Abstract”.

Line 38: A reference is needed.

Two references have been added.

Line 70: The reference Lagostena et al (doi: 10.1038/srep43900) should be included in this list.

The reference has been included.

General remark: It should be discussed how one can reconcile the fact that TPC1 and TPC2 KO mice have such a mild phenotype if TPC channels are so important for trafficking.

We agree with the reviewer that importance of TPCs for intracellular trafficking should be reflected in the phenotypes of existing knockout models. Therefore, we added the following paragraph in the “Introduction – Function in the endolysosomal system”:

In view of the importance of TPCs for intracellular trafficking one might expect more severe phenotypes in existing knockout models. However, a closer inspection of these mouse models suggests that phenotypes come to light mostly under stimulated, but not under basal conditions. This is evident in liver cells from TPC2 deficient mice, here, only mice fed with a high cholesterol diet demonstrated an accumulation of cholesterol in late endosomes probably due to an impairment of endolysosomal fusion processes. Furthermore, TPC2 deficient mice were susceptible for cholesterol overload and liver damage only under this diet (Grimm et al., 2014). Further example is the IgE-triggered exocytosis observed in mast cells (Arlt et al., 2020).

Still unpublished work in our lab supports this notion. We observe an important role of TPC1 for endocytosis, recycling or degradation of co-transporters in the renal proximal tubule. This function emerges not under basal conditions, but becomes evident only after hormonal stimulation or by challenging the acid-base balance. Please also note that besides TPCs other ion channels participate in regulation of endolysosomal trafficking and published TPC knockout models only lack one functional TPC gene, so that at least a partial functional compensation of the absent gene is plausible.

Round 2

Reviewer 1 Report

General comments:

The revision improved the manuscript. I still have some minor corrections listed below.

Specific comments:

  1. Line 28: Shorten “The diversity of tpc-genes, exon-intron structures, transcripts and proteins together with sequence comparisons and phylogenetic relationships were extensively described in basic publications” TO “ The diversity of tpc-genes, exon-intron structures, transcripts and proteins and phylogenetic relationships were recently described.”

  2. Line 114: Change “TPC2 deficient mice were susceptible for cholesterol overload and liver damage only under this diet” TO “TPC2 deficient mice were susceptible to cholesterol overload and liver damage only under this diet”.

  3. Line 119: Delete “above mentioned”.

  4. Line 129: Change “we take advantage of a detailed RNA sequencing” TO “we use RNA sequence”.

  5. Line 132: Delete “suitable”.

  6. Line 249: Delete “detailed”.

  7. Line 418: Change “They show a different distribution pattern with TPC1 predominantly found in early and recycling endosomes and TPC2 mainly expressed in late endosomes and lysosomes.” TO “TPC1 is predominantly found in early and recycling endosomes and TPC2 is mainly expressed in late endosomes and lysosomes.”

  8. Line 439: Change “further corroborating an important role of TPC for all forms of endocytosis” TO further corroborating an important role for TPC in different types of endocytosis”.

Author Response

We thank the reviewer for his/her comments and suggestions, in particular, we are very grateful for the ranking of our work in the general review report form! 

General comments: The revision improved the manuscript. I still have some minor corrections listed below.

Specific comments:

  1. Line 28: Shorten “The diversity of tpc-genes, exon-intron structures, transcripts and proteins together with sequence comparisons and phylogenetic relationships were extensively described in basic publications” TO “ The diversity of tpc-genes, exon-intron structures, transcripts and proteins and phylogenetic relationships were recently described.”
  2. Line 114: Change “TPC2 deficient mice were susceptible for cholesterol overload and liver damage only under this diet” TO “TPC2 deficient mice were susceptible to cholesterol overload and liver damage only under this diet”.
  3. Line 119: Delete “above mentioned”.
  4. Line 129: Change “we take advantage of a detailed RNA sequencing” TO “we use RNA sequence”.
  5. Line 132: Delete “suitable”.
  6. Line 249: Delete “detailed”.
  7. Line 418: Change “They show a different distribution pattern with TPC1 predominantly found in early and recycling endosomes and TPC2 mainly expressed in late endosomes and lysosomes.” TO “TPC1 is predominantly found in early and recycling endosomes and TPC2 is mainly expressed in late endosomes and lysosomes.”
  8. Line 439: Change “further corroborating an important role of TPC for all forms of endocytosis” TO “further corroborating an important role for TPC in different types of endocytosis”.

Thank you for drawing our attention to all these points, we corrected all the mistakes or rather changed the sentences accordingly. All changes are indicated in red in the manuscript file.